# Are young and older children with diarrhea presenting in the same way?

Sharika Nuzhat[1], Baharul Alam[1], S. M. Tafsir Hasan[2], Shamsun Nahar Shaima[2], Mohammod Jobayer Chisti[1], A. S. G. Faruque[2], Rina Das[2,3‡]*, Tahmeed Ahmed[2,4,5,6‡]

1 Clinical and Laboratory Services, International Centre for Diarrhoeal Disease Research, Bangladesh (icddr, b), Mohakhali, Dhaka, Bangladesh, 2 Nutrition Research Division, International Centre for Diarrhoeal Disease Research, Bangladesh (icddr,b), Mohakhali, Dhaka, Bangladesh, 3 Gangarosa Department of Environmental Health, Rollins School of Public Health, Emory University, GA, United States of America, 4 James P. Grant School of Public Health, BRAC University, Dhaka, Bangladesh, United States of America, 5 Department of Global Health, University of Washington, Seattle, WA, United States of America, 6 Office of the Executive Director, International Centre for Diarrhoeal Disease Research, Bangladesh (icddr,b), Mohakhali, Dhaka, Bangladesh

‡ RD and TA are contributed equally as senior author on this work.
* rina.das@icddrb.org, rina.das@emory.edu

**Data Availability Statement:** It is the policy of our centre (icddr,b) that the data, which contain identifying patient information, are not to be made available. However, data related to this paper are available upon request. Researchers who meet the

## Abstract

### Background

Diarrhoea is a global health problem. More than a quarter of diarrhoeal deaths occur among children less than five years. Different literatures analyzed presentation and outcomes of less than five diarrhoeal children. The world has made remarkable progress in reducing child mortality. So, older children are growing in number. Our aim was to identify clinical differentials and variations of pathogens among younger (less than five) and older (five to nine years) diarrhoeal children.

### Method

Data were extracted from the diarrhoeal disease surveillance system (DDSS) of Dhaka Hospital (urban site) and Matlab Hospital (rural site) of the International Centre for Diarrhoeal Disease Research, Bangladesh for the period of January 2012 to December 2021. Out of 28,781 and 12,499 surveillance patients in Dhaka and Matlab Hospital, 614 (2.13%) and 278 (2.22%) children were five to nine—years of age, respectively. Among under five children, 2456 from Dhaka hospital and 1112 from Matlab hospital were selected randomly for analysis (four times of five to nine years age children, 1:4).

### Results

Vomiting, abdominal pain, and dehydrating diarrhoea were significantly higher in older children in comparison to children of less than five years age (p-value <0.05) after adjusting study site, gender, antibiotic use before hospitalization, diarrhoeal duration < 24 hours, intake of oral rehydration fluid at home, parental education, WASH practice and history of cough. *Vibrio. cholerae*, *Salmonella*, and *Shigella* were the common fecal pathogen observed among older children compared to under five after adjusting for age, gender and study site.

criteria for access to confidential data may contact with Armana Ahmed (armana@icddrb.org) at the Research Administration of icddr,b (http://www.icddrb.org/).

**Funding:** The author(s) received no specific funding for this work.

**Competing interests:** The authors have declared that no competing interests exist.

## Conclusion

Although percentage of admitted diarrhoeal children with five to nine years is less than under five years children but they presented with critical illness with different diarrhoeal pathogens. These observations may help clinicians to formulate better case management strategies for children of five to nine years that may reduce morbidity.

## Introduction

Diarrhoea is a global health problem. In 2016, diarrhea was the eighth leading cause of mortality among the total population, responsible for more than 1·6 million deaths [1]. More than a quarter (26·93%) of diarrhoeal deaths occurred among children younger than five years, and about 90% (89·37%) of diarrhoeal deaths occurred in South Asia and sub-Saharan Africa [1]. A systematic review of diarrhoea morbidity and mortality [2], suggested that diarrhoea morbidity rates have remained constant in all ages since the 1980s in both developed and developing countries, also estimated more than 2.8 billion episodes of diarrhoea per year in children aged more than five years, adolescents, and adults [2]. Older children and adolescents are responsible for large burden of communicable disease [3].

Globally, diarrhoea is the second leading cause death among under five children [4]. In Asia, this risk remains high from ages 5–14 and reaches a low and constant plateau throughout adolescence and adulthood. At the same time, it declines less dramatically in Africa among more than five years old children and remains relatively stable throughout the lifespan [2]. In Bangladesh Demographic and Health Survey (BDHS) 2017–18, information was obtained from under five children about their experience of having an episode of diarrhoea in the two weeks before the survey. Overall, 5% of children under age of five years had diarrhoea during last two weeks period [5]. Prevalance remained the same in BDHS 2022 survey. But limited data were found on older children suffering from diarrhoea. Children in the age group of 5–14 years are regarded as school-age. This period lays the foundation for good health and sound mind in children, which persists throughout their lifetime [6]. The first growth spurt, the pre-adolescent or mid-growth spurt, is seen at around six to eight years of age. This is followed by the adolescent growth spurt between 10–17 years of age. There are many factors present in children's day-to-day household settings which make them vulnerable to undernutrition.

Infectious disease morbidity was associated with lower weight gains in Bangladeshi children aged 5 to 11 years [7]. Global diarrhoea mortality among individuals older than five years was dominated by *Shigella* [8] and nearly 70% occurred in children more than five years [8]. *Vibrio cholerae* (cholera) was the third leading cause of diarrhoea mortality among all ages, responsible for 0.11 million deaths [8]. In Bangladesh, diarrhoea mortality rates decreased among children under five from 15.1 to 6.0 per 1000 live births between 1980 and 2015 [9, 10].

In recent decades, little is known about the burden of diarrhoeal disease in children under five and in older children aged five to nine years. Hence, to address the existing knowledge gap and share research findings with policymakers and clinicians to formulate better case management strategies, we undertook this comparative assessment of the clinical features and etiology of diarrhoea among older children of five to nine years with that among under five years old children.

## Materials and methods

### Ethical statements

For this study, data were extracted from the electronic database of the hospital-based diarrhoeal disease surveillance system (DDSS) of Dhaka hospital and Matlab hospital of the International

Center for Diarrhoeal Disease Research, Bangladesh (icddr,b). The DDSS has the approval from the institutional review board of icddr,b (Research Review Committee and Ethical Review Committee) for data analysis. Ethical Review Committee was also pleased with the voluntary participation, maintenance of rights of the participants, and confidential handling of personal information by the hospital doctors and accepted this consenting procedure. At the time of enrolment into DDSS, verbal consent was obtained from the parents or the attending caregivers of each child, following hospital policy. The verbal consent was recorded by keeping a check-mark in the questionnaire that was again assured by showing the mark to parents or caregivers.

## Study population and study site

Diarrhoeal Disease Surveillance System (DDSS) is routine ongoing surveillance in hospitals of icddr,b located in Dhaka and Matlab, Bangladesh. In the Surveillance system of Dhaka Hospital, systematically (from every 50th patient according to their hospital ID number) collects information including age, sex, socio-demographic characteristics, clinical features, and identifies common bacterial and viral isolates from fecal samples. In Matlab Hospital, patients from the Matlab HDSS (Health and Demographic Surveillance System) area are included in the hospital surveillance system.

For the present study, we used 10 years data limited to 0–9 years of children enrolled in the DDSS from January 2012 to December 2021. In this study period, 28,781 and 12,499 diarrhoeal patients were enrolled in Dhaka and Matlab Hospital surveillance systems, respectively. Of them, 614 and 278 children aged five to nine years from Dhaka and Matlab, separately, were included in the analyses (Fig 1). We considered randomly selected children aged less than 5 years as the comparison group (controls) in a 1:4 ratio to increase the statistical power for analyses using SPSS [11], Dhaka Hospital was considered an urban study site, and Matlab Hospital was a rural study site.

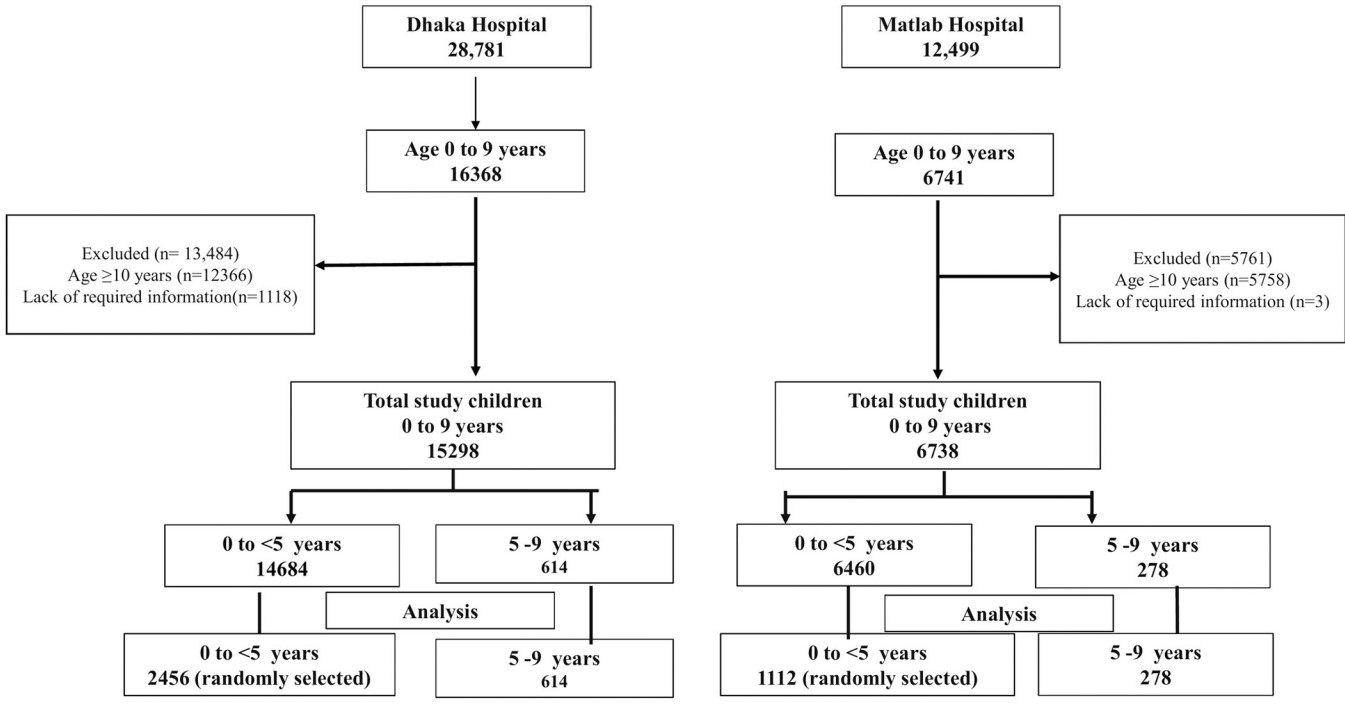

**Fig 1. Selection of study group.**

## Study design

In this retrospective study, five to nine years of diarrhoeal children were considered as cases and compared with under five diarrhoeal children. The study period was from 2012 to 2021. Fig 1 shows the flow chart for selecting the cases and comparison groups.

614, five to nine years old children and 2456 under-five children were selected from Dhaka hospital. 278, five to nine years children and 1112 under-five children were selected from Matlab hospital.

## Stool microbiology

After collection, fresh stool specimens from DDSS-enrolled patients were sent to icddr,b central laboratories for routine screening of the aforementioned enteric pathogens. The details of the laboratory procedures for detecting entero-pathogens in stool samples have been described elsewhere [12–14]. In brief, *Vibrio cholera* was isolated by growth on tellurite taurocholate gelatin agar (TTGA) media with enrichment in bile peptone broth. *Salmonella* spp. and *Shigella* spp. were isolated by growth on MacConkey agar and Salmonella-Shigella (SS) agar with enrichment in selenite broth followed by antisera panel testing (Denka Seiken Co., Ltd.). *Campylobacter* spp. was isolated by growth on Brucella agar. *Aeromonas* spp. were isolated by growth on TTGA and gelatin agar followed by phenotypic characterization of long-sugar metabolism. For the detection of ETEC, fresh stool specimens were plated onto MacConkey agar. The plates were incubated at 37˚C for 18 hours. Six lactose fermenting individual colonies morphologically resembling E. coli were isolated and tested for the presence of heat-stable and heat-labile toxins using ganglioside GM1 ELISA and multiplex PCR [15, 16]. Antimicrobial susceptibility was checked for each of the isolated bacteria. The presence of Group, A rotavirus-specific VP6 antigen in stool samples, was detected using the ProSpect Rotavirus kit (Oxoid Ltd, Basingstoke, UK), which utilizes a polyclonal antibody in a solid phase sandwich-type enzyme immunoassay according to the manufacturer's instructions [17].

## Statistical analysis

We summarized the characteristics of children using percentages and mean with standard deviation (SD) as appropriate. Bivariable and multivariable binary logistic regression models were used to assess the association of outcome variables (dehydration status, presented with fever, presence of watery stool, abdominal pain, convulsion, death, and infection with rotavirus, *Vibrio. Cholerae*, *Salmonella*, *and Shigella*) with age groups (five to nine years compared to less than five years). The number of observations in one cell was less than five for the convulsion and death variable, so we could not include these two variables in the final model. Each of the clinical outcome variables was analyzed as a separate model adjusting for potential available covariates, such as study site, age, sex, use of antibiotics prior hospitalization, diarrhoeal duration, use of oral rehydration fluid at home, parent's education, WASH practice, and history of cough. Isolated pathogens were adjusted for study site, age, and sex. We expressed the strength of association as odds ratio (OR) and adjusted odds ratio (aOR) with a 95% confidence interval (95% CI) with a <0.05 p value. All statistical tests were two-sided. Data analysis were done in Stata v15.1 (Stata Corp, College Station, TX, USA).

## Results

In Dhaka hospital, there were 28,781 surveillance patients during the period 2012 to 2021. Out of them 614, five to nine years old children and 2456, under five children were included in analysis. (Fig 1). From 2012 to 2021in Matlab Hospital total surveillance patients were 12,499.

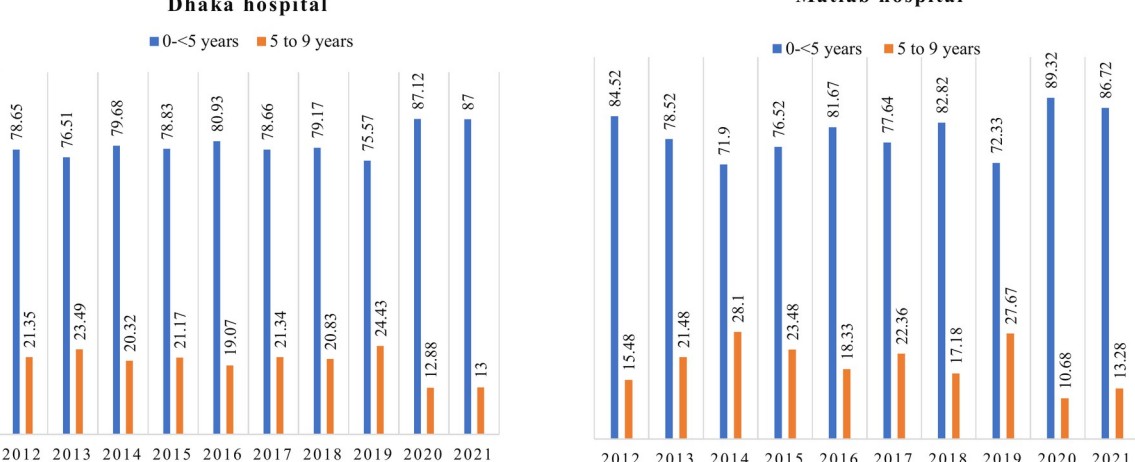

**Fig 2. Yearly admission percentage of five to nine years and under five years children in Dhaka hospital and Matlab hospital 2012–2021.**

Among them, we included 278, five to nine years old children and 1112 under five children for analysis (Fig 1). Yearly admission of the children of both groups in both study sites was comparable though it showed the rise of younger children during the COVID pandemic 2020–2021. In the COVID period, older children presenting with diarrhea were lower in percentage than younger children (Fig 2).

Tables 1 and 2 show the comparison of demographic and clinical characteristics of five to nine years children and less than five years children. Parents of older children were illiterate compared to younger children in both study areas. Five to nine years children of urban and rural sites more commonly presented with short duration of diarrhoea (<24 hours), some/ severe dehydration, vomiting, and abdominal pain. On the other hand, these children were less likely to present with a history of fever and cough. No death was observed among the older children in both hospitals.

Table 3 shows that in logistic regression analysis after adjusting for study site, sex, use of antibiotics before hospitalization, diarrhoeal duration, use of fluid at home, parent's education, source of drinking water, use of water treatment, use of improved latrine, h/o cough, five to nine years old diarrhoeal children were significantly associated with some /severe dehydration, vomiting, and abdominal pain.

Table 4 shows isolated organisms among the study group in both the study sites. All microorganisms were not tested in the rural study site. Isolation of *V. Cholerae*, *Shigella*, and *Salmonella* was significantly higher in the study areas among five to nine years children than under five children. On the other hand, Rotavirus predominance was observed among the young children group. After adjusting the study site, age, and sex, *V. Cholerae*, *Shigella*, and *Salmonella* were significantly associated with five to nine years children than under-five children. At the same time, rotavirus was substantially lower in older children.

## Discussion

To our knowledge, this is the first study that investigated the clinical presentation and etiology of diarrhoea among five to nine years old children of Bangladesh and compared it with younger children. Though the percentage of five to nine years diarrhoeal children has been limited

**Table 1. Characteristics of five to nine years and under five diarrhoeal children admitted in urban site (Dhaka hospital), icddr,b from 2012–2021.**

| | | 5–9 years (n = 614) (%) | 0- <5 years (n = 2456) (%) |
|---|---|---|---|
| **Sex** | Female | 246 (40.07) | 944 (38.44) |
| | Male | 368 (59.93) | 1512 (61.56) |
| **Paternal education** | Literate | 446 (72.64) | 2143/2455 (87.29) |
| | Illiterate | 168 (27.36) | 312/2455 (12.71) |
| **Maternal education** | Literate | 483 (78.66) | 2265/2455 (92.26) |
| | Illiterate | 131(21.34) | 190/2455 (7.74) |
| **Sources of drinking water** | Non-tube well | 378 (61.56) | 1254 (51.08) |
| | Tube well water | 236 (38.44) | 1201/2455 (48.92) |
| **Water treatment method used** | No | 343/613 (55.95) | 1475 (60.06) |
| | Yes | 270/613 (44.05) | 981 (39.94) |
| **Toilet facility** | Non-sanitary latrine | 78 (12.70) | 303 (12.34) |
| | Sanitary/semi-sanitary latrine | 536 (87.30) | 2152/2455 (87.66) |
| **Use of oral rehydration fluid at home** | Yes | 592 (96.42) | 2369/2455 (96.50) |
| | No | 22 (3.58) | 86/2455 (3.50) |
| **Use of Antibiotic prior to hospitalization** | No | 232/366 (63.39) | 578/1454 (39.75) |
| | Yes | 134/366 (36.61) | 876/1454 (60.25) |
| **Diarrheal duration < 24 hours** | No | 259 (42.18) | 1689 (68.80) |
| | Yes | 355 (57.82) | 766/2455 (31.20) |
| **Stool consistency** | Non-watery | 43 (7.00) | 208 (8.47) |
| | Watery | 571 (93.00) | 2247/2455 (91.53) |
| **Presence of vomiting** | No | 104 (16.94) | 673 (25.94) |
| | Yes | 510 (83.06) | 1819 (74.06) |
| **Presence of abdominal Pain** | No | 220 (35.83) | 1317 (53.65) |
| | Yes | 394 (64.17) | 1138/2455 (46.35) |
| **Dehydration status** | No | 124 (20.20) | 1673/2454 (68.17) |
| | Some | 240 (39.09) | 699 (28.48) |
| | Severe | 250 (40.72) | 82/2454 (3.34) |
| **Presence of fever** | No | 397/594 (66.84) | 1430 (60.21) |
| | Yes | 197/594 (33.16) | 945/2375 (39.79) |
| **Presence of cough** | No | 500 (81.43) | 1426 (58.09) |
| | Yes | 114 (18.57) | 1029/2455 (41.91) |
| **Presence of convulsion** | No | 588 (95.77) | 2454/2455 (99.96) |
| | Yes | 26(4.23) | 1/2455 (0.40) |
| **Outcome** | Alive | 614 (100.00) | 2449/2453 (99.84) |
| | Death | 0 | 4/2453 (0.31) |

in number in the last 10 years in both urban and rural sites, these older children presented with critical illnesses.

The world has made remarkable progress in reducing child mortality. The global U5MR decreased by 59% (90% uncertainty interval [UI] 56–61) from 93·0 (91·7–94·5) deaths per 1000 live births in 1990 to 37·7 (36·1–40·8) in 2019 [18]. If all countries met the SDG target on under five mortality, 11 million under five deaths could be averted between 2020 and 2030 [18]. With more young children now surviving, improving the survival of older children is an increasing area of focus nowadays. Globally, mortality rates among older children declined by 40% during this period [19]. Every region of the world has made progress in reducing more older child mortality rates, with the most significant percentage reductions occurring in

**Table 2. Characteristics of 5–9 years and under five diarrhoeal children admitted in rural site (Matlab hospital), icddr,b from 2012–2021.**

| characteristics | | 5–9 years (n = 278) (%) | 0-<5 years (n = 1112) (%) |
|---|---|---|---|
| **Sex** | Female | 107 (38.49) | 450 (40.47) |
| | Male | 171 (61.51) | 662 (59.53) |
| **Paternal education** | Literate | 236 (84.89) | 1022 (91.99) |
| | Illiterate | 42 (15.11) | 89/1111 (8.01) |
| **Maternal education** | Literate | 247 (88.85) | 1,073 (96.49) |
| | Illiterate | 31 (11.15) | 39 (3.51) |
| **Sources of drinking water** | Non-tube well | 4 (1.44) | 36 (3.24) |
| | Tube well water | 274 (98.56) | 1076 (96.76) |
| **Water treatment method used** | No | 269 (96.76) | 1,025 (92.18) |
| | Yes | 9 (3.24) | 87 (7.82) |
| **Toilet facility** | Non-sanitary latrine | 186 (66.91) | 711 (63.94) |
| | Sanitary/semi-sanitary latrine | 92 (33.09) | 401 (36.06) |
| **Use of oral rehydration fluid at home** | Yes | 248 (89.21) | 964 (86.69) |
| | No | 30 (10.79) | 148 (13.31) |
| **Use of Antibiotic prior to hospitalization** | No | 107/193 (55.44) | 347/723 (47.99) |
| | Yes | 86/193 (44.56) | 376/723 (52.01) |
| **Diarrheal duration < 24 hours** | No | 151 (54.32) | 790 (71.04) |
| | Yes | 127 (45.68) | 322 (28.96) |
| **Stool consistency** | Non-watery | 65 (23.38) | 211 (18.97) |
| | Watery | 213 (76.62) | 901 (81.03) |
| **Presence of vomiting** | No | 48 (17.27) | 278 (25.00) |
| | Yes | 230 (82.73) | 834 (75.00) |
| **Presence of abdominal Pain** | No | 60 (21.58) | 553 (49.73) |
| | Yes | 218 (78.42) | 559 (50.27) |
| **Dehydration status** | No | 203 (73.02) | 1064 (95.68) |
| | Some | 68 (24.46) | 45 (4.05) |
| | Severe | 7 (2.52) | 3 (0.27) |
| **Presence of fever** | No | 70/240 (29.17) | 186/1000 (18.60) |
| | Yes | 170/240 (70.83) | 814/1000 (81.40) |
| **Presence of cough** | No | 234 (84.17) | 668 (60.07) |
| | Yes | 44 (15.83) | 444 (39.93) |
| **Presence of convulsion** | No | 99/100 (99.00) | 450/453 (99.34) |
| | Yes | 1/100 (1.00) | 3/453 (0.66) |
| **Outcome** | Alive | 277 (100.00) | 1112 (100.00) |
| | Death | 0 | 0 |

**Table 3. Association of clinical characteristics with children aged five to nine years in logistic regression.**

| | Unadjusted OR (95%CI) | P value | Adjusted OR (95%CI) | P value |
|---|---|---|---|---|
| **Some /severe dehydration** | 5.70 (4.88–6.67) | **0.000** | 6.50 (5.07–8.35) | **0.000** |
| **Fever** | 0.72 (0.62–0.84) | **0.000** | 0.73 (0.58–0.92) | **0.007** |
| **Watery stool** | 0.97 (0.77–1.21) | 0.765 | 0.86 (0.63–1.16) | 0.325 |
| **Presence of vomiting** | 1.68 (1.39–2.03) | **0.000** | 1.55 (1.21–1.99) | **0.000** |
| **Abdominal Pain** | 2.41 (2.06–2.81) | **0.000** | 2.60 (2.10–3.21) | **0.000** |

Each model was adjusted for study site, sex, use of antibiotics before hospitalization, diarrhoeal duration, use of fluid at home, parents education, source of drinking water, use water treatment, use of improved latrine, and history of cough seperately.

**Table 4. Isolated fecal organisms in diarrhoeal children admitted in urban site (Dhaka hospital) and rural site (Matlab hospital), icddr,b from 2012–2021 and independent association of organisms with five to nine years group.**

| Organisms | Dhaka Hospital | | | | Matlab hospital | | | | Overall | |
| --- | --- | --- | --- | --- | --- | --- | --- | --- | --- | --- |
| | 5–9 years (n = 239) | 0-<5 years (n = 1104) | Crude OR (95% CI) | P Value | 5–9 years (n = 109) | 0-<5 years (n = 469) | Crude OR (95% CI) | P Value | Study site and sex adjusted OR (95% CI) | P value |
| Rotavirus | 45/239 (18.83) | 838/1104 (75.91) | 0.07 (0.05–0.10) | 0.000 | 13/109 (11.93) | 397/469 (84.65) | 0.02 (0.01–0.05) | 0.000 | 0.05 (0.04–0.07) | 0.000 |
| ETEC | 13/236 (5.51) | 80/1103 (7.25) | 0.75 (0.41–1.36) | 0.339 | Not done | Not done | | | | |
| *V. Cholerae* | 103/239 (43.10) | 39/1104 (3.53) | 20.68 (13.73–31.14) | 0.000 | 34/109 (31.19) | 14/469 (2.99) | 14.73 (7.55–28.75) | 0.000 | 18.93 (13.34–26.86) | 0.000 |
| *Shigella* | 22/239 (9.21) | 25/1104 (2.26) | 4.38 (2.42–7.90) | 0.000 | 52/109 (47.71) | 50/469 (10.66) | 7.64 (94.75–12.31) | 0.000 | 6.14 (4.24–8.89) | 0.000 |
| *Campylobacter* | 14/239 (5.86) | 57/1104 (5.16) | 1.14 (0.63–2.09) | 0.663 | No isolates | No isolates | | | | |
| *Aeromonas* | 29/239 (12.13) | 56/1104 (5.07) | 2.58 (1.61–4.14) | 0.000 | No isolates | No isolates | | | | |
| *Salmonella* | 7/239 (2.93) | 8/1104 (0.72) | 4.13 (1.48–11.51) | 0.003 | 10/109 (9.17) | 8/469 (1.71) | 5.82 (2.24–15.12) | 0.000 | 4.97 (2.48–9.97) | 0.000 |

aOR: adjusted odds ratio, each of the pathogens was adjusted for study site, and sex

Southeast Asia (65%), Africa (60%), and European (40%) LMICs (lower and middle income countries) [19]. In 2016, about 1 million children died, mainly from preventable causes like pneumonia, diarrhoea and malaria [20]. Globally, the top five causes of death among older children were diarrhoeal diseases (10%), lower respiratory infections (10%), road traffic injuries (8%), malaria (7%), and meningitis (6%) in 2016 [19]. Despite absolute reductions in mortality rates, African LMICs had substantially higher older child mortality rates than all other regions due to the continued burden of infectious diseases, including diarrhoeal diseases, lower respiratory infections, and malaria [19]. Southeast Asian LMICs, in contrast, had the fastest progress of any region. This success was mainly achieved by reducing mortality rates from the exact infectious causes that continue to burden the African continent [19].

The focus on the diarrhoeal disease was on under-five children. In our study, we have found that older children are more likely to be associated with dehydrating diarrhoea, vomiting, and abdominal pain in the adjusted model compared to younger children. Several studies have assessed the accuracy of the WHO, CDS, and DHAKA methods of dehydration assessment in different contexts, none have been validated for assessing dehydration in patients over five years with acute diarrhoea [21–24]. Differences in both adult physiology and diarrhoea etiology may compromise the accuracy of clinical diagnostic models initially developed for use in young children [25, 26]. NIRUDAK is the first study to empirically derive clinical diagnostic models for assessing dehydration severity in patients over five years of age [27]. A systematic review reported the estimates of duration and severity outcomes [28]. Due to the lack of studies reporting results for children 5–15 years of age, they could not directly estimate the burden of diarrhoea on this age group [28]. Dehydration in younger children is associated with poor intake, and amount of purging in comparison to low body weight [29], but for older children lack of knowledge regarding ORS intake, negligence probably the factor associated with their presentation. A study conducted on determinants of dehydrating diarrhoea reported that risk of severe dehydration in diarrhoea increases abruptly from zero to ten years of age and *Vibrio cholerae* is identified as contributing organism for dehydration [12]. But this study did not differentiate other clinical features between young and older children.

Childhood morbidity status, especially diarrhoea, has been reported in other studies to impair the growth of children, specifically in weight gain [30] which in the long run halts the development of the children. A study conducted in a neighboring country found a significant association between being underweight and episodes of diarrhoea in the last year among school-aged children [31]. This follows the findings of Torres et al. [7], who reported that diarrhoea correlates significantly with retarded weight gain among children above preschool age. So, diarrhoeal illness among these older children needs to be appropriately treated, and proper nutrition counseling is also required for post-diarrhoeal days.

Another study conducted for older children, adolescents, and adults revealed that the most frequently isolated pathogens among patients hospitalized for diarrhoea are ETEC and *Vibrio cholerae* O1/O139 [25]. Rotavirus, which is known to be a leading cause of death among young children, was not found to be as important among older persons providing additional evidence suggesting immunity with increasing age [25]. In outpatient settings, *Salmonella* spp., *Shigella* spp., and *E. histolytica* were the most frequently isolated pathogens in the older age group [25]. In our study, we analyzed our hospitalized diarrhoeal children where rotaviral diarrhoea is less likely associated with more than five years of children. This is similar to other studies suggesting better protection with older age [32]. But *Salmonella* spp. and *Shigella* spp. were significantly higher in percentage among older age group compared to younger children.

Irrespective of the scientific pertinence of the abovementioned findings on five to nine years children, our study has some limitations. We included only the diarrhoeal children who visited the hospital. Lack of data related to caregivers' s knowledge regarding the management of diarrhoeal along with data regarding follow-up of such children was not available. We could not evaluate the effect of the Rota viral vaccine as this is yet to incorporate into the EPI schedule of Bangladesh. Although this study had been conducted in the largest diarrhoeal disease facility in the world and also focused on both urban and rural communities, such data as mentioned above along with nationwide data on the disease severity and fatality among older children would have enhanced the reliability of our observation. In addition to that the areas where the burden of *Cholera* is insignificant might show different presentation.

## Conclusion

Our study suggested that children over five might get benefit from greater attention to controlling diarrhoeal diseases, thereby reducing their morbidity. International initiatives targeting child mortality have focused on children under five years of age [7]. Including older children in the program, management would help to reduce morbidity and mortality in this age group who are neglected. High-quality prospective community- and facility-based studies on the etiology of diarrhoeal illness among children aged five to nine years old are needed, ideally, ones that cover an entire 12-month period to account for diarrhoea seasonality.

## Acknowledgments

This research was supported by core donors who provide unrestricted support to icddr,b for its operations and research. Current donors providing unrestricted support include the Governments of Bangladesh and Canada. We gratefully acknowledge our core donors for their support and commitment to icddr,b's research efforts.

## Author Contributions

**Conceptualization:** Sharika Nuzhat, A. S. G. Faruque, Rina Das, Tahmeed Ahmed.

**Formal analysis:** Sharika Nuzhat, S. M. Tafsir Hasan, Rina Das.

**Methodology:** Sharika Nuzhat.

**Supervision:** Rina Das, Tahmeed Ahmed.

**Writing – original draft:** Sharika Nuzhat.

**Writing – review & editing:** Baharul Alam, S. M. Tafsir Hasan, Shamsun Nahar Shaima, Mohammod Jobayer Chisti, A. S. G. Faruque, Rina Das, Tahmeed Ahmed.

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
