## [Decision Letter · Decision Letter 0]

8 Dec 2023

PONE-D-23-25038Are young and older children with diarrhea presenting in the same way?PLOS ONE

Dear Dr. Das.

Thank you for submitting your manuscript to PLOS ONE. After careful consideration, we feel that it has merit but does not fully meet PLOS ONE’s publication criteria as it currently stands. Therefore, we invite you to submit a revised version of the manuscript that addresses the points raised during the review process.

We look forward to receiving your revised manuscript.

Kind regards,

Sanjoy Kumer Dey, M.D

Academic Editor

PLOS ONE

Journal Requirements:

Did you know that depositing data in a repository is associated with up to a 25% citation advantage (https://doi.org/10.1371/journal.pone.0230416)? If you’ve not already done so, consider depositing your raw data in a repository to ensure your work is read, appreciated and cited by the largest possible audience. You’ll also earn an Accessible Data icon on your published paper if you deposit your data in any participating repository (https://plos.org/open-science/open-data/#accessible-data).

The name of the colleague or the details of the professional service that edited your manuscript.A copy of your manuscript showing your changes by either highlighting them or using track changes (uploaded as a *supporting information* file).A clean copy of the edited manuscript (uploaded as the new *manuscript* file).

"This research was funded by core donors who provide unrestricted support to icddr,b for its operations and research. Current donors providing unrestricted support include the Governments of Bangladesh and Canada. We gratefully acknowledge our core donors for their support and commitment to icddr, b's research efforts."

Reviewers' comments:

Reviewer's Responses to Questions

**Comments to the Author**

1. Is the manuscript technically sound, and do the data support the conclusions?

Reviewer #1: Yes

Reviewer #2: Yes

2. Has the statistical analysis been performed appropriately and rigorously? 

Reviewer #1: Yes

Reviewer #2: No

3. Have the authors made all data underlying the findings in their manuscript fully available?

Reviewer #1: No

Reviewer #2: Yes

4. Is the manuscript presented in an intelligible fashion and written in standard English?

Reviewer #1: Yes

Reviewer #2: No

5. Review Comments to the Author

Reviewer #1: This is an understudied and important topic given most research on diarrhea has focused on the under-5 group in LMICs. This is a straightforward but well done dataset analysis of older children at two hospitals in bangladesh that helps provide an understanding of the presentation and epidemiology of this older group for those working in the region.

General comments:

- how was the 5-9 age group decided to be the focus of this study as presumably other age groups (particularly adolescents) were part of the datasets? Adolescents are another highly understudied group, and one might consider including these patients in analysis, or discuss whether there may be plans to include in a future study which would be greatly needed

- suggest that the authors revise the discussion for improved clarity and flow, to focus on how this study's results fits in with the prior literature and give recommendations on how the authors would like to see their results applied to improve patient care and public health policies. additionally what are the highest priority areas for research in this older child group that are needed?

- suggest plos one team assist with english grammar due to minor errors throughout - for instance

"5-9 years of children" should be rephrased to "children 5-9 years of age" etc

specific edits:

Line 55-7: please cite the mortality rates from diarrhea in older children globally as well as in bangladesh/asian region if these exist

Study population/study site:

- Please include more details on inclusion/exclusion criteria. Also how was diarrhea defined? Did this include acute/chronic diarrhea? What about other disease in which diarrhea was just one presenting symptom?

- Clarify how random selection for controls was performed, what were inclusion/exclusion criteria for the 0-5 year group?

- Please also give some more details about icddrb and the Dhaka and Matlab hospitals for readers who are unaware. what types of patients are served, differences between matlab/dhaka, cholera seasonality, etc.

- Do patients with the most critical illness get transferred? I note that it is later mentioned there were no deaths in this age group at either hospital - is this due to transfer for more severe cases to other facilities perhaps? Would give some context and data if available, or cite as a limitation if not available.

Statistical analysis

- how were the outcome variable and covariates selected from those available in the datasets? Please provide rationale and citations whenever possible

Line 151: what was suspected reason for demographic change during covid? How would this impact interpretation of your results?

Line 161: a the authors expand on their speculations why these variables (parental illiteracy, shorter duration, more dehydration, etc) were associated with the older child group? shorter duration perhaps to higher presence of cholera/less rotavirus? Or other explanation? Can add this in discussion. also how might these findings be used to improve care by clinicians who treat these patients?

Line 171: h/o should be spelled out

Line 216: spell out abbreviations such as CDS at first use

Discussion section:

- In general this section needs to be more focused and to emphasize the most interesting findings of THIS study. there is relatively little discussion of the actual findings from the current study. additionally there is quite a bit of discussion on general child mortality however this should be reframe to focus on diarrheal mortality and how this study findings fit into the existing literature.

- There is no comment on the findings of those variables associated with older groups - parental literacy, short duration of symptoms, cough, fever, convulsions, etc. these are interesting findings and would like the authors to expand on why these were associated in this patient population. Particularly how to frame the understanding of how parental literacy and diarrheal presentation in older children are associated in Bangladesh context.

Line 221-22 - need more details about this review, what it studied, the main findings. Details are lacking and hard to understand how this current study results fit with this

Reviewer #2: Comments

Major comments:

1) Your manuscript requires major revision of % writing

2) First of all, your outcome (dependent variable) should be “diarrhea” for your Table 1 and Table 2. Therefore, you have to re-construct Table 1 and Table 2 using ‘Diarrhea’ as outcome variable and all the variables listed in both tables as independent variables using bivariable logistic regression to calculate the unadjusted odds ratio. Later on, those variables socring p-value < 0.25 to take them to the final model (Multivariable logistic regression).

3) Your Table 3 is incorrect. You have to construct another Table 3 which mainly focuses on multivariable logistic regression by taking those variables from Table 1 and Table 2 (after incorporating the comments given in the above 2nd major comments) which scored p-values less than or equal to 0.25. Mind you, you have to analyses both unadjusted odds ratio and adjusted odds ratio in Table 3 after including those variables with a score of p-value < 0.25. Therefore, all your Tables (Table 1, Table 2, and Table 3) are incorrect. With these incorrect analyses you cannot, discuss, you cannot conclude and you cannot recommend. Therefore, I will stop reviewing your manuscript now. This is because, all your results, discussions, conclusion and recommendations will definitely be changed after complying with the comments given above. Mind you, this is your major of major revisions. Therefore, I will review your second manuscript after incorporating all what has been commented above.

Minor comments:

1) Avoid unnecessary open space between title and authors name and between authors name and affiliation. Example, line number 2 and 5 have nothing to indicate

2) Your Table 4 may be correct, but I will see it after you incorporated all the major comments given above.

Sections

Page-2

Abstract

Rewrite your abstract using the structured way of abstract writing i.e. divide it into sub-sections Background, Method, Results, and Conclusion

L25: every single digit number should be written in word. Therefore, write <5 in word “less than five” and do the same correction throughout the document

L26: delete “children” and replace it by “diseases”

L26 – L27: the sentence “More young children are now surviving and improving the survival of older children” is meaningless. Re-write it meaningfully!!!

L33: Every time the word “respectively” should be preceded by comma. Therefore, add comma after the word “age”

L38: Don’t start a sentence with abbreviation; rather write in full form as “Vibrio cholerae”

Page-3

Introduction

L75 – L76: The justification “In recent decades, little is known about the burden of diarrheal disease in older children and adolescents” doesn’t go in line with your title. Therefore, modify it as “In recent decades, little is known about the burden of diarrheal disease in children under five and in older children five to nine years”

Page-4

L81: add ‘s’ to statement and make it “Statements”

L86: Don’t start a sentence with abbreviation ‘ERC’. Rather, write the full form ‘Ethical Review Committee’

Page-5

L93: Same comment as above, write the full form of DDSS

L104: the 11th reference should be placed between SPSS and the full stop

L108: Delete ‘0-<5’ and replace it by ‘under five’

Page-6

L110 – L111: Delete all the (%) from the two lines

Page-7

L133: Delete ‘simple’ and replace it by ‘bivariable’

L136: Delete ‘< 0-5 years’ and replace it by ‘less than five years’

L146: Delete colon from ‘Result:’

L147 – L148: the sentence “From 2012 to 2021, in Dhaka Hospital, out of a total of 28,781 surveillance patients, 5 to 9 years of children were 614 (%), randomly selected 2456 (%) children from 14,684 under -5 children (Figure 1).” is non-sense. Please re-write it meaningfully

L149 – L151: still the sentence “During the study period, in Matlab Hospital, out of 12,499 surveillance patients, 5 to 9 years children were 278 (%), and we randomly selected 1112 (%) children from 6,460 under-five children (Fig 1).” is non-sense. Re-write it meaningfully

Page-8

L160: Don’t start a sentence with number. Write it as “Five to nine years”

L164 – L 166: From Table 1 delete ‘Ref’ and replace it by ‘1’. This is because you are taking that option as a reference, so it will have an Odds Ratio of 1. Furthermore, italicize CI and p from p-value. Similarly, do the same correction throughout the manuscript.

In this Table 1 please enclose the CIs of the variables ‘The water treatment method used’ and ‘Toilet facility’. Moreover, where are the ORs with the CI for the variables ‘Presence of convulsion’ and ‘Outcome’?

From Table 2 for the variable ‘Maternal education’ you have to take the option ‘Literate’ as a reference; therefore, delete the OR with its CI and replace it by 1

Comments on References

Reference #3: It is too old data in 2008, it doesn’t describe the current situation. Therefore, remove it and replace it by the most recent one

Reference #4: It is too old data in 2008, it doesn’t describe the current situation. Therefore, remove it and replace it by the most recent one

Reference #9: Lacks page number. Therefore, include the page number after going back to the original article

I will continue evaluating the references in your second draft after addressing the major comments given above!!!

6. PLOS authors have the option to publish the peer review history of their article (what does this mean?). If published, this will include your full peer review and any attached files.

Reviewer #1: No

Reviewer #2: No

---

## [Author Response · Author response to Decision Letter 0]

16 Jan 2024

Reviewer's comments to the authors:

Major comments:

Comment:

1) Your manuscript requires major revision of % writing 

Response: Thank you for your comment. We have revised the write up.

Comment:

2) First of all, your outcome (dependent variable) should be “diarrhea” for your Table 1 and Table 2. Therefore, you have to re-construct Table 1 and Table 2 using ‘Diarrhea’ as outcome variable and all the variables listed in both tables as independent variables using bivariable logistic regression to calculate the unadjusted odds ratio. Later on, those variables socring p-value < 0.25 to take them to the final model (Multivariable logistic regression). 

Response: Thank you very much for your suggestion. As our hospital is a diarrheal disease hospital all patients were with diarrhea. There is no scope to consider diarrhea as outcome variable. So, we compared the prevalence of different clinical presentation between two age group of population in two different study sites (Urban-Dhaka hospital and Rural-Matlab hospital) (Table 1 and Table 2). Later we have analyzed each of the clinical features of diarrhea which are our outcome variables (dehydration status, presented with fever, presence of watery stool, abdominal pain). Each outcome variable was analyzed considering with separate models (Table 3) with similar adjusted variables as our target is to find out the differentiating feature for older children for early detection of ill cases.

3) Your Table 3 is incorrect. You have to construct another Table 3 which mainly focuses on multivariable logistic regression by taking those variables from Table 1 and Table 2 (after incorporating the comments given in the above 2nd major comments) which scored p-values less than or equal to 0.25. Mind you, you have to analyses both unadjusted odds ratio and adjusted odds ratio in Table 3 after including those variables with a score of p-value < 0.25. Therefore, all your Tables (Table 1, Table 2, and Table 3) are incorrect. With these incorrect analyses you cannot, discuss, you cannot conclude and you cannot recommend. Therefore, I will stop reviewing your manuscript now. This is because, all your results, discussions, conclusion and recommendations will definitely be changed after complying with the comments given above. Mind you, this is your major of major revisions. Therefore, I will review your second manuscript after incorporating all what has been commented above. 

Response: Thank you for your comments. All of our study children are with diarrhea, so there is no scope to consider diarrhea as outcome variable. In table 1 and table 2, we compared the prevalence of different demographic and clinical presentation between two age group of population in two different study sites (Urban-Dhaka hospital and Rural-Matlab hospital). We have removed OR, CI and p value for table 1 and 2 as all of those are not our outcome variables. In table 3, we have analyzed our outcome variables (dehydration status, presented with fever, presence of watery stool, abdominal pain). Each outcome variable was analyzed considering as separate model (Table 3) with similar adjusted variables as our target is to find out the differentiating feature for older children for early detection of ill cases. 

Minor comments: 

Comment:

1) Avoid unnecessary open space between title and authors name and between authors name and affiliation. Example, line number 2 and 5 have nothing to indicate

Response: Thank you, we have reduced the spaces in title page.

Comment:

2) Your Table 4 may be correct, but I will see it after you incorporated all the major comments given above.

Response: Thank you for your comment. As our outcome variables for table 4 are organisms, so we have included both crude and adjusted OR for all the organisms in different sites. As all of our children are with diarrhea culture positive organisms were included in table 4. 

Sections

Page-2

Abstract

Comment: Rewrite your abstract using the structured way of abstract writing i.e. divide it into sub-sections Background, Method, Results, and Conclusion

Response: Thank you for your suggestion. We organized the abstract as per journal policy. We have added sub-headings as per your suggestion.

Comment: L25: every single digit number should be written in word. Therefore, write <5 in word “less than five” and do the same correction throughout the document

Response: Thank you for your comment. We have revised single digits with texts. 

Comment: L26: delete “children” and replace it by “diseases”

Response: We revised this.

Comment: L26 – L27: the sentence “More young children are now surviving and improving the survival of older children” is meaningless. Re-write it meaningfully!!!

Response: Thank you for your comment. We have rewritten the line. 

Comment: L33: Every time the word “respectively” should be preceded by comma. Therefore, add comma after the word “age”

Response: Thank you. We have included comma after “age”.

Comment: L38: Don’t start a sentence with abbreviation; rather write in full form as “Vibrio cholerae”

Response: Thank you for your suggestion. We have revised it accordingly.

Page-3

Introduction

Comment: L75 – L76: The justification “In recent decades, little is known about the burden of diarrheal disease in older children and adolescents” doesn’t go in line with your title. Therefore, modify it as “In recent decades, little is known about the burden of diarrheal disease in children under five and in older children five to nine years”

Response: Thank you for your suggestion. We have revised it accordingly

Page-4

Comment: L81: add ‘s’ to statement and make it “Statements”

Response: We have included “s” for “statement”.

Comment: L86: Don’t start a sentence with abbreviation ‘ERC’. Rather, write the full form ‘Ethical Review Committee’

Response: We have incorporated “Ethical Review Committee” in place of ERC.

Page-5

Comment: L93: Same comment as above, write the full form of DDSS

Response: Thank you. We have incorporated elaborated form.

Comment: L104: the 11th reference should be placed between SPSS and the full stop

Response: We have incorporated full stop after reference.

Comment: L108: Delete ‘0-<5’ and replace it by ‘under five’

Response: Thank you for your comment. We have revised it.

Page-6

Comment: L110 – L111: Delete all the (%) from the two lines

Response: Thank you. We have removed these.

Page-7

Comment: L133: Delete ‘simple’ and replace it by ‘bivariable’

Response: we have revised it.

Comment: L136: Delete ‘< 0-5 years’ and replace it by ‘less than five years’

Response: We have revised it as text. 

Comment: L146: Delete colon from ‘Result:’ 

Response: Thank you for your comment, we have removed colon.

Comment: L147 – L148: the sentence “From 2012 to 2021, in Dhaka Hospital, out of a total of 28,781 surveillance patients, 5 to 9 years of children were 614 (%), randomly selected 2456 (%) children from 14,684 under -5 children (Figure 1).” is non-sense. Please re-write it meaningfully

Response: Thank you for your suggestion. We have revised the sentence. 

Comment: L149 – L151: still the sentence “During the study period, in Matlab Hospital, out of 12,499 surveillance patients, 5 to 9 years children were 278 (%), and we randomly selected 1112 (%) children from 6,460 under-five children (Fig 1).” is non-sense. Re-write it meaningfully.

Response: Thank you. We have revised the sentence.

Page-8

Comment: L160: Don’t start a sentence with number. Write it as “Five to nine years”

Response: We have revised the number with texts.

Comment: L164 – L 166: From Table 1 delete ‘Ref’ and replace it by ‘1’. This is because you are taking that option as a reference, so it will have an Odds Ratio of 1. Furthermore, italicize CI and p from p-value. Similarly, do the same correction throughout the manuscript. 

Response: Thank you for your comment. We have removed OR and CI and p value from both table 1 and 2 as all of these are not our outcome variables, these are derailing readers insight. 

Comment: In this Table 1 please enclose the CIs of the variables ‘The water treatment method used’ and ‘Toilet facility’. Moreover, where are the ORs with the CI for the variables ‘Presence of convulsion’ and ‘Outcome’?

Response: Thank you for your comment. We have removed OR and CI and p value from both table 1 and 2 as all of these are not our outcome variables, these are derailing readers insight. 

Comment: From Table 2 for the variable ‘Maternal education’ you have to take the option ‘Literate’ as a reference; therefore, delete the OR with its CI and replace it by 1

Response: Thank you for your comment. We have removed OR and CI and p value from both table 1 and 2 as all of these are not our outcome variables, these are derailing readers insight. 

Comments on References

Comment: Reference #3: It is too old data in 2008, it doesn’t describe the current situation. Therefore, remove it and replace it by the most recent one

Response: Thank you for your suggestion. We have replaced the reference with the article published in 2023.

Comment: Reference #4: It is too old data in 2008, it doesn’t describe the current situation. Therefore, remove it and replace it by the most recent one

Response: Thank you for your observation. We have replaced the reference with the article published in 2023. 

Comment: Reference #9: Lacks page number. Therefore, include the page number after going back to the original article

Response: Thank you for your observation. We have revised the reference.

Comment:

I will continue evaluating the references in your second draft after addressing the major comments given above!!!

Response: Thank you for your kind support.

Comments to the author:

In this instance, it seems there may be acceptable restrictions in place that prevent the public sharing of your minimal data. However, in line with our goal of ensuring long-term data availability to all interested researchers, PLOS’ Data Policy states that authors cannot be the sole named individuals responsible for ensuring data access (http://journals.plos.org/plosone/s/data-availability#loc-acceptable-data-sharing-methods).

Data requests to a non-author institutional point of contact, such as a data access or ethics committee, helps guarantee long term stability and availability of data. Providing interested researchers with a durable point of contact ensures data will be accessible even if an author changes email addresses, institutions, or becomes unavailable to answer requests. Before we proceed with your manuscript, please also provide non-author contact information (phone/email/hyperlink) for a data access committee, ethics committee, or other institutional body to which data requests may be sent. If no institutional body is available to respond to requests for your minimal data, please consider if there any institutional representatives who did not collaborate in the study, and are not listed as authors on the manuscript, who would be able to hold the data and respond to external requests for data access? If so, please provide their contact information (i.e., email address). Please also provide details on how you will ensure persistent or long-term data storage and availability.

Response:

Thank you for your valuable comment. In the methodology section of the manuscript, we added the data availability statement. We also provide non-author contact information from icddr,b Research Administration as “Data Availability: This data set contains some personal information of the study patients (such as name, admission date, month, area of residence). Our IRB has required that the personal information of the participants is not disclosed. Thus, the policy of our centre (icddr,b) is that we should not make the availability of the whole data set in the manuscript, the supplemental files, or a public repository. However, data related to this manuscript are available upon request and researchers who meet the criteria for access to confidential data may contact Armana Ahmed (armana@icddrb.org) to the Research Administration of icddr,b (http://www.icddrb.org/).”

---

## [Decision Letter · Decision Letter 1]

7 Mar 2024

Are young and older children with diarrhea presenting in the same way?

PONE-D-23-25038R1

Dear Rina Das

We’re pleased to inform you that your manuscript has been judged scientifically suitable for publication and will be formally accepted for publication once it meets all outstanding technical requirements.

Kind regards,

Sanjoy Kumer Dey, M.D

Academic Editor

PLOS ONE

Additional Editor Comments (optional):

Reviewers' comments:

Reviewer's Responses to Questions

**Comments to the Author**

1. If the authors have adequately addressed your comments raised in a previous round of review and you feel that this manuscript is now acceptable for publication, you may indicate that here to bypass the “Comments to the Author” section, enter your conflict of interest statement in the “Confidential to Editor” section, and submit your "Accept" recommendation.

Reviewer #3: All comments have been addressed

2. Is the manuscript technically sound, and do the data support the conclusions?

Reviewer #3: Partly

3. Has the statistical analysis been performed appropriately and rigorously? 

Reviewer #3: Yes

4. Have the authors made all data underlying the findings in their manuscript fully available?

Reviewer #3: Yes

5. Is the manuscript presented in an intelligible fashion and written in standard English?

Reviewer #3: Yes

6. Review Comments to the Author

Reviewer #3: The authors have adequately addressed all the comments raised in a previous round of review. I am completely satisfied with their responses. I think, the manuscript is suitable to publish in this journal.

7. PLOS authors have the option to publish the peer review history of their article (what does this mean?). If published, this will include your full peer review and any attached files.

Reviewer #3: **Yes: **Shuvra Kanti Dey

---

## [Editor Report · Acceptance letter]

30 Apr 2024

PONE-D-23-25038R1 

PLOS ONE

Dear Dr. Das, 

I'm pleased to inform you that your manuscript has been deemed suitable for publication in PLOS ONE. Congratulations! Your manuscript is now being handed over to our production team.

Kind regards, 

on behalf of

Dr. Sanjoy Kumer Dey 

Academic Editor

PLOS ONE